# Role of transoesophageal echocardiography in detecting patent foramen ovale in stroke patients aged ≤60 years: A retrospective study

**Reabal Najjar**[1]*, **Andrew Hughes**[1,2]

1 Australian National University Medical School, Australian National University, Acton, ACT, Australia,
2 Department of Neurology, The Canberra Hospital, Garran, ACT, Australia

* Reabal.Najjar@anu.edu.au

## Abstract

### Background

The underlying aetiology of ischaemic strokes is unknown in as many as 50% of cases. Patent foramen ovale (PFO) has become an increasingly recognised cause of ischaemic strokes in young patients. The present study aimed (1) to assess the frequency of transoesophageal echocardiography (TOE) performed and the proportion of PFOs detected in patients aged ≤60 years and (2) examine the effect of PFO closure on reducing stroke reoccurrence.

### Methods

This was a retrospective clinical audit based on de-identified, secure medical records of the Canberra Hospital, Australia. A review of records was conducted on discharged patients aged 18–60 years admitted to the stroke unit following an ischaemic stroke episode between January 1, 2015, and December 31, 2018.

### Results

A total of 214 acute ischaemic stroke patients were admitted to the stroke unit (mean age, 49.2 ± 9.7 years). Concerning aetiology, 47.2% were cryptogenic in origin, whereas 52.8% had a stroke of a determined cause. 12 patients were diagnosed with a PFO and 7 venous thromboembolic events were identified, 1 in the cryptogenic group and 6 in the determined cause group. 91.7% of PFOs were diagnosed in patients with a cryptogenic stroke. Transthoracic echocardiography (TTE) was performed in 37.3% of patients and had detected 4 PFOs (sensitivity 27.3%, specificity 92.5%). TOE was performed in 26.2% of patients and had detected 11 PFOs (sensitivity 90.0%, specificity 100%). The number needed to treat to prevent the occurrence of an ischaemic stroke through PFO closure was estimated at 30.

### Conclusions

An inverse association between age and PFO presence was found in patients aged 18–60 years. Additionally, TOE was superior to TTE for detecting PFO, particularly in those with

**Data Availability Statement:** The data is part of the inpatient stroke database of the Canberra Hospital and cannot be shared publicly as the dataset contains potentially identifying and sensitive

information that would compromise patient privacy. De-identified data access may be made available upon reasonable request by contacting the ACT Health Human Research Ethics Committee (HREC) email address at ethics@act.gov.au.

**Funding:** The author(s) received no specific funding for this work.

**Competing interests:** The authors have declared that no competing interests exist.

stroke of an undetermined cause. Our results suggest an increased need for TOE as a routine imaging procedure for acute ischaemic stroke patients aged ≤60.

## Introduction

The causes of acute ischaemic stroke cannot be identified in up to 40% of cases [1]. Further, 3% of strokes occur in patients aged <40 years; with no underlying aetiology found in roughly half of these patients [2]. According to the Australian Bureau of Statistics, stroke is the third leading cause of mortality in Australia, with 10,869 stroke fatalities in 2015, accounting for 6.8% of the 159,052 deaths [3]. Although young adults comprise only 10% of overall stroke patients, stroke is nevertheless a major cause of long-term disability with a disproportionately heavy emotional and socioeconomic impact on the young population, leaving many patients disabled during their most productive years [4].

Recent studies have shown that patent foramen ovale (PFO) is an increasingly established cause of undetermined or cryptogenic ischaemic strokes, particularly in young patients with no other identifiable stroke risk factors [3]. PFO is a congenital cardiac lesion that persists into adulthood and is considered the most prevalent defect of the atrial septum of the heart and most common cardiac cause of stroke in young patients, with paradoxical embolism being the presumed cause of stroke [4, 5]. Accordingly, PFO is thought to be an important cause of ischaemic strokes and is an independent risk factor for stroke [6]. PFO is present in some form in 25% of the general population and may be detected through transthoracic echocardiography (TTE) or transoesophageal echocardiography (TOE) [7]. In stroke patients aged <45 years, the prevalence of PFO is up to 50% [8].

Until recently, the evidence supportive of PFO closure in patients presenting with cryptogenic stroke was lacking. In 2012 and 2013, there were 3 negative trials on PFO closure, all of which showed a lack of evidence for PFO closure [9–11]. Four positive studies on PFO closure emerged in 2017 and 2018, showing a benefit of PFO closure in reducing the long-term stroke risk [12–14]. A recent meta-analysis of 6 randomised controlled trials (RCTs) of a new trial published in March 2018 concluded that PFO closure significantly lowered stroke reoccurrence rate in patients with high-risk PFO characteristics [15]. These subsequent positive trials demonstrated the benefit of closure, which was achieved by adopting a better device design and by carefully selecting patients and following them for a longer period to reduce future stroke occurrence [16].

We aimed to assess the number of patients aged ≤60 years, presenting with an acute ischemic stroke, who might benefit from PFO closure. The study aimed to identify the frequency of TTE and TOE performed and determine whether such imaging procedures should be performed routinely on patients presenting with an ischaemic stroke in this age group. Moreover, we aimed to identify what proportion of patients in whom PFO was detected would fit the criteria from the RCTs for closure to reduce the frequency of future strokes.

## Materials and methods

### Study design

A retrospective clinical audit was conducted to identify patients aged ≤60 years admitted to the Canberra Hospital acute stroke unit between January 1, 2015, and December 31, 2018 with an ischaemic stroke episode. All patients presenting with an ischaemic stroke were admitted to the stroke unit and were identified from the stroke databases of the hospital.

The study was submitted to and approved by the ACT Health Human Research Ethics Committee's Low-Risk Sub-Committee (2018.LRE.00213). Following ethical clearance, digitised patient record documents stored in the centralised Clinical Record Information System of the Canberra Hospital were examined. To maintain anonymity, all patient clinical documents were de-identified and were only referenced by an allocated unique numerical identifier.

As this was a retrospective descriptive study, there was no intervention or change in treatment for the participants. Additionally, informed consent was not required as patients were not subjected to any treatment.

Sources of information included the attending physician's patient notes, discharge summaries, and cardiac and neuroimaging results (computed tomography [CT], magnetic resonance imaging [MRI], TTE, and TOE). Relevant patient documents were reviewed and subsequently collated and analysed using an Excel spreadsheet (Microsoft Corp., Redmond, WA).

## Patients

The study population comprised 214 acute ischaemic patients out of 323 patients admitted to the acute stroke unit (Fig 1). As this review aimed to discern the role of TOE in ischaemic stroke in young patients, we only included patients aged 18–60 years. In line with previous RCTs, we defined 'young adults' as those younger than 60 years [9–14, 17].

Patients were included in the review if they were aged ≥18 years and ≤60 years, presented with an ischaemic stroke, and had been admitted to the acute stroke unit. We excluded 109 patients who presented with a diagnosis other than ischaemic stroke (i.e. haemorrhagic stroke, transient ischaemic attack, or other neurological deficits), had inadequate data, or were discharged against medical advice (Fig 1).

Stroke was clinically diagnosed by the caring neurologist based on the revision of relevant patient documents and was based on clinical presentation with a recent onset of a neurological deficit for 24 hours and cerebrovascular imaging results. The primary neuroimaging procedures used in the diagnosis of ischaemic stroke were non-contrast CT and diffusion-weighted MRI [18].

The cause of stroke was classified according to the TOAST (Trial of ORG 10172 in acute stroke treatment) classification [19]. The TOAST classification system includes 5 categories: 1) large-artery atherosclerosis, 2) cardioembolism, 3) small-artery occlusion (lacune), 4) stroke of other determined cause, and 5) stroke of undetermined cause [19]. For this study, stroke categories were further classified into either stroke of a determined cause or stroke of an undetermined cause (cryptogenic). An ischaemic stroke was identified as cryptogenic after all other causes of stroke were eliminated and no identifiable mechanism was established. To assess whether atherosclerosis and cardioembolism, or small artery occlusion were potential causes of stroke, neurological imaging results from CT angiography, MRI, and magnetic resonance angiography records were reviewed. Patients were classified according to whether they underwent TTE or TOE imaging and whether it demonstrated a PFO.

Although TTE is regarded as the mainstay for diagnostic cardiac ultrasound due to its less invasive and less expensive nature in comparison to TOE, the latter is considered a beneficial complementary imaging modality. TOE provides greater visualisation of posterior cardiac structures without the interference produced by lung, bone, and thoracic wall. However, as a result of its semi-invasive design, sedation involvement, as well as high training requirements, the potential benefits of conducting TOE must outweigh the risks associated with the procedure. Therefore, in certain clinical cases, TTE may precede TOE in order to preclude or guide TOE. In this centre, 91.1% (51/56) of patients who underwent TOE had received TTE prior as

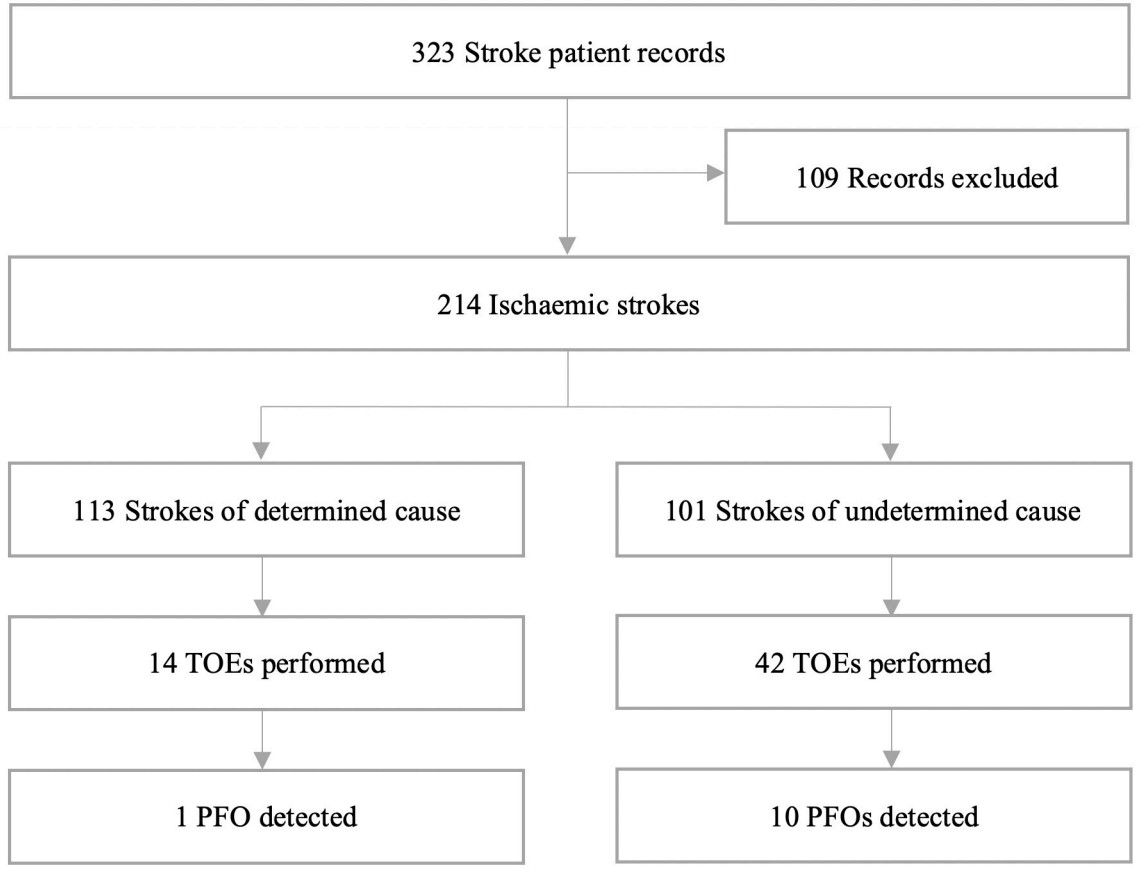

**Fig 1. Flow diagram of selection criteria.** PFO, patent foramen ovale; TOE, transoesophageal echocardiography.

the latter was either a technically challenging study or produced poor quality images. The remaining 8.9% who underwent TOE without TTE had a high degree of suspicion for PFO and no common risk factors for stroke.

## Statistical analysis

Quantitative parameters were expressed as mean ± standard deviation, whereas qualitative parameters were presented as frequencies and percentages. Statistical significance for intergroup differences was tested using two-sided unpaired Student's t-test for continuous variables and the Mann–Whitney test for non-normal variables. Categorical variables were compared using the chi-square test ($p < 0.05$ for statistical significance). TTE and TOE were compared using sensitivity, specificity, positive predictive values, negative predictive values, and likelihood ratios. The number needed to treat was calculated from 7 randomised trials for risk estimates where risk difference was significant. Statistical analysis of the data was performed using the software package IBM-SPSS version 26.0 (IBM Corp., Armonk, NY). All patient data were entered into Microsoft Excel 2019 (Microsoft Corp., Redmond, WA).

## Results

Currently, the Canberra Hospital admits 285 patients with stroke to the acute stroke unit annually. Of these patients, approximately 22% are aged ≤60 years. In all, 323 patients aged

**Table 1. Patient demographics and baseline characteristics.**

| | Cryptogenic (n = 101) | Determined cause (n = 113) | All patients (n = 214) |
|---|---|---|---|
| **Sex—no. of patients (%)** | | | |
| Male | 59 (58.4) | 72 (63.7) | 131 (61.2) |
| Female | 42 (41.6) | 41 (36.3) | 83 (38.8) |
| **Age—no. of patients (%)** | | | |
| 18–30 years | 8 (7.9) | 6 (5.3) | 14 (6.5) |
| 31–40 years | 14 (13.9) | 8 (7.1) | 22 (10.3) |
| 41–50 years | 32 (31.7) | 27 (23.9) | 59 (27.6) |
| 51–60 years | 47 (46.5) | 72 (63.7) | 119 (55.6) |
| Average age (years) | 47.2 ± 9.9 | 51 ± 9.1 | 49.2 ± 9.7 |
| **Medical history—no. of patients (%)** | | | |
| Hypertension | 31 (30.7) | 64 (56.6) | 95 (44.4) |
| Hyperlipidaemia | 28 (27.7) | 44 (38.9) | 72 (33.6) |
| Diabetes mellitus | 18 (17.8) | 27 (23.9) | 45 (21.0) |
| Smoking status | | | |
| Current | 17 (16.8) | 29 (25.7) | 46 (21.5) |
| Past | 13 (12.9) | 14 (12.4) | 27 (12.6) |
| Previous stroke/TIA | 9 (8.9) | 18 (15.9) | 27 (12.6) |
| Deep vein thrombosis | 1 (1.0) | 6 (5.3) | 7 (3.3) |
| Migraine | 6 (5.9) | 5 (4.4) | 11 (5.1) |
| IHD/CAD | 8 (7.9) | 34 (30.1) | 42 (19.6) |
| Septal defect | 2 (2.0) | 0 (0.0) | 2 (0.9) |
| PFO | 12 (11.9) | 1 (0.9) | 13 (6.1) |

TIA, transient ischaemic attack; IHD, ischaemic heart disease; CAD, coronary artery disease; PFO, patent foramen ovale.

18–60 years were admitted to the stroke unit of the Canberra Hospital between January 1, 2015, and December 31, 2018.

## Baseline characteristics

A total of 214 records were used in this retrospective audit. The age of admitted patients ranged from 18 to 60 years (mean age, 49.2 ± 9.7 years). Of the 214 patients, 14 (6.5%) were aged ≤30 years, 22 (10.3%) were between the ages of 31 and 40 years, 59 (27.6%) were between the ages of 41 and 50 years, and 119 (55.6%) were aged ≥51 years. Further, 131 (61.2%) patients were male, whereas 83 (38.8%) were female (Table 1).

The cause of the stroke was identifiable from the clinical record in 113 (52.8%) patients. In the remaining 101 (47.2%) patients, the stroke was categorised as cryptogenic. The average age of patients with cryptogenic stroke was 47.2 years, whereas the average age of patients with stroke with a determined cause was 51 years. Therefore, on average, cryptogenic stroke patients were 4 years younger (Table 2).

## Risk factors

The common risk factors identified were hypertension, hyperlipidaemia, type 1 and 2 diabetes mellitus, and current/past smoking. Patients presenting with cryptogenic stroke were less likely to present with the aforementioned risk factors for stroke than patients with stroke of determined cause (Table 2). Compared with patients with cryptogenic stroke, patients with a stroke of determined cause had more than twice the rate of hypertension, almost twice the rate

**Table 2. Imaging and PFO detection rates.**

| | MRI (n = 140) | TTE (n = 81) | TOE (n = 56) | Patients with PFO (n = 12) | PFO detected with TTE (n = 4)[a] | PFO with detected TOE (n = 11)[a] | All (n = 214) |
|---|---|---|---|---|---|---|---|
| **Sex—no. of patients (%)** | | | | | | | |
| Male | 77 (55.0) | 44 (55.0) | 27 (48.2) | 7 (58.3) | 3 (75.0) | 7 (63.6) | 131 (61.2) |
| Female | 63 (45.0) | 37 (45.7) | 29 (51.8) | 5 (41.7) | 1 (25.0) | 4 (36.4) | 83 (38.8) |
| **Age—no. of patients (%)** | | | | | | | |
| 18–30 years | 13 (9.3) | 13 (16.3) | 11 (19.6) | 4 (33.3) | 2 (50.0) | 4 (36.4) | 14 (6.5) |
| 31–40 years | 17 (12.1) | 21 (25.9) | 15 (26.8) | 2 (16.7) | 0 (0.0) | 2 (18.2) | 22 (10.3) |
| 41–50 years | 39 (27.9) | 21 (26.3) | 17 (30.4) | 4 (33.3) | 2 (50.0) | 3 (27.3) | 59 (27.6) |
| 51–60 years | 71 (50.7) | 26 (32.5) | 13 (23.2) | 2 (16.7) | 0 (0.0) | 2 (18.2) | 119 (55.6) |
| **Cause of stroke—no. of patients (%)** | | | | | | | |
| Stroke of undetermined cause | 72 (51.4) | 51 (63.0) | 42 (75.0) | 11 (91.7) | 4 (100) | 10 (90.9) | 101 (47.2) |
| Stroke of determined cause | 68 (48.6) | 26 (32.5) | 14 (25.0) | 1 (8.3) | 0 (0.0) | 1 (9.1) | 113 (52.8) |
| Large artery atherosclerosis | 14 (10.0) | 4 (5.0) | 1 (1.8) | 0 (0.0) | 0 (0.0) | 0 (0.0) | 24 (11.2) |
| Cardioembolic | 51 (36.4) | 26 (32.5) | 13 (23.2) | 1 (8.3) | 0 (0.0) | 1 (9.1) | 84 (39.3) |
| Small artery occlusion (lacune) | 2 (1.4) | 0 (0.0) | 0 (0.0) | 0 (0.0) | 0 (0.0) | 0 (0.0) | 2 (0.9) |
| Stroke of other determined aetiology | 1 (0.7) | 0 (0.0) | 0 (0.0) | 0 (0.0) | 0 (0.0) | 0 (0.0) | 3 (1.4) |

MRI, magnetic resonance imaging; TTE, transthoracic echocardiography; TOE, transoesophageal echocardiography; PFO, patent foramen ovale.

[a]Three PFOs were diagnosed via both TOE and TTE.

of hyperlipidaemia and diabetes, and more than 4 times the rate of ischaemic heart disease/coronary artery disease. More importantly, 11 patients produced a positive bubble study in the cryptogenic group through TTE and TOE. Conversely, only 1 PFO was detected in the non-cryptogenic group (Table 2).

## TTE and TOE usage rates

The main imaging modalities used were CT scan, MRI, TTE, and TOE (Table 2). The cause of stroke had a significant influence on TTE and TOE usage rates. Patients presenting with cryptogenic stroke showed a significantly higher TTE and TOE administration rate than patients presenting with a stroke of determined cause. TTE usage rates were 50.5% in cryptogenic patients in comparison to 23.0% in patients with stroke of determined cause, and the TOE usage rates were 41.6% in cryptogenic patients in comparison to 12.4% in patients with stroke of determined cause ($p < 0.05$). Moreover, the TTE and TOE usage rates correlated inversely with patient age. In this study, 92.9% of patients aged 18–30 years underwent TTE and 78.6% underwent TOE, while only 21.8% and 10.9% of patients aged 51–60 years underwent TTE and TOE, respectively ($p < 0.05$). With regards to sex, while many more male patients (n = 131) were included in this clinical audit compared to female patients (n = 83), female patients had higher TTE (44.6%) and TOE usage rates (34.9%) than male patients (33.6% and 20.6%, respectively).

## Association with PFO

Overall, 12 patients were diagnosed with PFO. TTE detected PFO in 4 patients in the cryptogenic group, whereas TOE detected PFO in 10 patients (3 patients were diagnosed with PFO via both TTE and TOE). In contrast, only 1 PFO was detected in the group with determined stroke cause. The presence of PFO was much more common in patients presenting with

cryptogenic stroke as opposed to stroke of determined cause ($p < 0.05$). Similar to TTE and TOE administration patterns, the presence of PFO was also inversely associated with age. Younger stroke patients aged 18–30 years had a PFO detection rate of 28.6%. In contrast, only 1.7% of older stroke patients aged 51–60 years were found to have a PFO ($p < 0.05$).

## Diagnostic performance

Sensitivity, specificity, and predictive values were determined from the population of patients who had undergone both TTE and TOE (n = 51). Among the 51 patients, TTE demonstrated a positive bubble study in 3 patients but failed to detect PFO in 8 patients, yielding a sensitivity of 27.3%. TTE correctly identified 37 without PFO and falsely identified 3 patients without PFO, thus having a specificity of 92.5%. TOE demonstrated a positive bubble study in 9 cases but failed to detect 1 PFO, yielding a sensitivity of 90.0%. TOE also correctly identified all patients without PFO, leading to a sensitivity of 100%. The sensitivities, specificities, predictive values, and likelihood ratios of both imaging modalities are shown in Table 3.

## Discussion

### Main findings

Two significant associations were found in this clinical audit. First, there was a significant association between age and the presence of PFO. Previous studies reported that the prevalence of PFO ranged from 20 to 25% in the general population and might reach up to 56% in patients aged <55 years who had experienced a cryptogenic stroke [20–22]. With increasing age, the prevalence of PFO detection decreases linearly [20, 23]. In the present study, there was a higher prevalence of PFO among younger patients than among older stroke patients. Additionally, TOE was superior to TTE in detecting and excluding a PFO. These findings can be partially explained by higher detection rates observed in this age group through increased TTE and TOE administration rates and may not necessarily be due to younger age.

Second, we found a strong association between age and the number of TTE and TOE performed. In patients aged 18–30, 92.9% underwent TTE and 78.6% underwent TOE, while 21.8% and 10.9% of patients aged 51–60 underwent TTE and TOE, respectively. TOE is considered the gold standard for detecting a PFO. In our study, TOE detected 11 PFOs in contrast to 4 PFOs detected through TTE [24–26]. TOE is superior to TTE and is considered the imaging technique of choice regarding PFO diagnosis in patients presenting with a cryptogenic stroke due to its ability to delineate the morphological features of a PFO [26]. Consequently, in order to address the hypothesis that TOE is warranted as a routine investigation in stroke patients aged ≤60 years, we examined the relationship between the age of patients presenting with a cryptogenic stroke and TOE usage rates.

An inverse association was discovered whereby the younger a stroke patient was, the higher the likelihood that they would have a TOE ordered; our results showed a two-fold increase in TTE administration and a three-fold increase in TOE administration in patients presenting

**Table 3. Diagnostic performance data of TTE and TOE for PFO detection.**

| Imaging modality | Sensitivity (95% CI) | Specificity (95% CI) | PPV (95% CI) | NPV (95% CI) | +LR | -LR |
|---|---|---|---|---|---|---|
| TTE | 27.3% (0.097 to 0.566) | 92.5% (0.801 to 0.974) | 50.0% (0.188 to 0.812) | 82.2% (0.687 to 0.907) | 3.64 | 0.79 |
| TOE | 90.0% (0.596 to 0.982) | 100% (0.914 to 1) | 100% (0.701 to 1) | 97.6% (0.877 to 0.996) | ∞ | 0.1 |

TTE, transthoracic echocardiography; TOE, transoesophageal echocardiography; PPV, positive predictive value; NPV, negative predictive value; +LR, positive likelihood ratio; -LR, negative likelihood ratio; CI, confidence intervals.

with cryptogenic stroke ($p < 0.05$). 30 patients who received a negative TTE result did not go on to have a TOE performed, most likely due to the belief of the treating neurologist regarding the lack of benefit of PFO closure based on the original negative trials.

This audit found that older patients, specifically those within the 51–60-year age range, were more likely to experience a stroke with a determined cause associated with common risk factors, whereas younger patients in the 18–30-year age range were more likely to experience a cryptogenic stroke. This difference in stroke aetiology could likely explain the different rates of TOE being ordered by treating neurologists. Based on the extrapolation of our data, if the remaining cryptogenic stroke patients (n = 59) had a TOE ordered, then 11 additional PFOs would have potentially been detected.

In summary, we found that TTE and TOE were more likely to be performed in younger patients. Additionally, the younger the patient, the more likely that a PFO was detected. Whether this was a true effect or rather, a reflection of increased test frequency in younger adults remains uncertain.

## Previous studies

Despite the strong association between PFO and cryptogenic strokes in younger patients, 3 early RCTs did not report statistically significant differences in stroke rates for PFO closure and medical treatment [9–11].

The CLOSURE I trial in 2012, RESPECT trial in 2013, and PC trial in 2013 all showed negative results as they found no significant benefit of PFO closure over conservative management with respect to the future prevention of recurrent strokes [9–11]. These 3 negative studies have not demonstrated the superiority of PFO closure over medical treatment primarily due to the broad inclusion criteria that allowed the addition of patients with ischaemic strokes who would not benefit from PFO closure, such as those with non-cryptogenic strokes and a relatively short follow-up period.

A meta-analysis was subsequently conducted in March 2016 on individual patient data from the previous 3 negative trials in order to standardise the outcome definitions and assess heterogeneity of treatment effects across patient groups. The analysis found a statistically significant association between PFO closure with a percutaneously implanted device and a reduction in recurrent stroke rates [27].

Three RCTs in 2017 showed generally positive findings regarding the superiority of PFO closure over medical treatment [12–14]. The trials Gore REDUCE, CLOSE, and RESPECT extended follow-up were similar to the 3 negative trials with respect to mean age, mean follow-up time, and treatment/closure group; however, they differed in their inclusion criteria. The latter trials demonstrated moderate benefit of PFO closure and were likely positive due to stricter entry criteria, leading to the inclusion of patients whose presenting stroke was secondary to PFO rather than another primary cause, such as cerebral artery stenosis, thromboembolism, or atrial fibrillation.

An updated meta-analysis was published in October 2018 of all 6 RCTs and a recently published randomised trial—DEFENSE-PFO [17, 28]. This newer RCT published in May 2018 also concluded that the rate of stroke recurrence was significantly lower with PFO closure combined with medication than with medication treatment alone [17]. The pooled analysis of data from all seven aforementioned RCTs in the meta-analysis demonstrated a 59% significant risk reduction in stroke reoccurrence in the closure group compared to the treatment only group [28].

The lack of evidence prior to the emergence of the 2017–2018 trials might explain the low TTE and TOE uptake found in our study, which covered the period of 2015–2018, as attending

neurologists were possibly less likely to order a TOE if PFO closure was inconsequential with regards to future stroke prevention. Accordingly, we examined the patient results of these recent four positive trials in order to extrapolate the data to fulfil the criteria for PFO closure to prevent future stroke rates.

### Closure criteria

Based on the 2017 and 2018 trial results, by restricting the characteristics for inclusion, it was demonstrated that PFO closure might potentially reduce the rate of recurrent ischaemic stroke compared to medical treatment and anticoagulation [12–14]. The trials only included patients with a PFO associated with presentations and symptoms of a cryptogenic stroke. Secondly, they reduced the possibility of alternate causes of ischaemic stroke apart from PFO by using a standardised assessment tool to evaluate previous cryptogenic stroke [29, 30]. Finally, the trials enrolled patients with a lower cardiovascular factor burden; therefore, there was a lower likelihood that strokes were caused by typical cardiovascular risk factors [31].

By applying such parameters, the average number needed to treat based on the trials was 30 persons. In other words, the number of patients in a population with similar characteristics to our sample of patients who would need to be treated with PFO closure to prevent 1 ischaemic stroke for 5 years is approximately 30 patients, estimated from the 3 most recent positive trials [12–14].

### Future proposals

Based on our results, there was considerable underutilisation of TTE and TOE in our population, particularly among young stroke patients. We hoped to redress this issue by justifying an increase in TOE uptake by physicians to reduce recurrent stroke rates through increased PFO detection. Extrapolation of our data concluded that an additional 11 PFOs would potentially have been detected if the remaining cryptogenic stroke patients (n = 59) had had a TOE ordered.

For future directions, there are few data on the cause-effect relationship of paradoxical embolism and cryptogenic stroke, which need to be addressed. Moreover, TOE in this study had been used as an adjunct test to TTE routinely whenever TTE produced suboptimal images. Therefore, to answer the question of whether TTE or TOE is superior with regards to PFO detection, future studies require the usage of TTE and TOE in all patients. Finally, future studies would need to assess the number of patients requiring a TOE and how many may benefit from PFO closure.

### Limitations

There are several limitations in this study, particularly owing to its retrospective nature and the biases inherent to its design. First, the data were collected from medical records, imaging results, and clinical documentation. Thus, reliance on data from previous records makes a more objective assessment challenging. Second, the low frequency of deep vein thrombosis (3.3%) in our sample size in comparison to the number of PFO found (5.6%) provides limited utility in assigning causality to PFO without identification of paradoxical embolism. However, the actual cause-effect relationship of paradoxical embolism is uncertain since the mechanisms of cryptogenic stroke have not yet been fully understood [32]. Third, selection bias was unavoidable as the decision to whether a patient received TTE or TOE was made by the caring neurologist responsible for the patient and, consequently, was not randomised. The final limitation of this study is that it is based on data from a single centre as well as the relatively small sample size given the magnitude and aims of the study.

## Conclusions

In conclusion, our study showed an inverse association between age and TOE uptake rates and between age and PFO presence in young stroke patients aged <60 years. Additionally, we found that TOE is considered the most sensitive test for detecting a PFO; therefore, TOE should be under consideration for use as a routine procedure in patients younger than 60 years of age presenting with a stroke of undetermined origin. The use of a more sensitive imaging modality among young ischaemic stroke patients will potentially decrease the future occurrence of paradoxical emboli associated with PFO.

## Author Contributions

**Conceptualization:** Andrew Hughes.

**Data curation:** Reabal Najjar, Andrew Hughes.

**Formal analysis:** Reabal Najjar, Andrew Hughes.

**Investigation:** Reabal Najjar.

**Methodology:** Reabal Najjar.

**Project administration:** Andrew Hughes.

**Software:** Reabal Najjar.

**Supervision:** Andrew Hughes.

**Validation:** Reabal Najjar, Andrew Hughes.

**Visualization:** Andrew Hughes.

**Writing – original draft:** Reabal Najjar.

**Writing – review & editing:** Reabal Najjar.

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
