## [Decision Letter · Decision Letter 0]

17 Aug 2020

PONE-D-20-21340

Role of transoesophageal echocardiography in detecting patent foramen ovale in young stroke patients: A retrospective study.

PLOS ONE

Dear Dr. Najjar,

Thank you for submitting your manuscript to PLOS ONE. After careful consideration, we feel that it has merit but does not fully meet PLOS ONE’s publication criteria as it currently stands. Therefore, we invite you to submit a revised version of the manuscript that addresses the points raised during the review process.

Reviewer comments support the necessity to improve data analysis, in term of explaining the bias of a retrospective analysis, instead of finding causality, as the characteristics of the present study could not bring a strong explanatory hypothesis due to its nature.

We look forward to receiving your revised manuscript.

Kind regards,

Miguel A. Barboza, MD, MSc

Academic Editor

PLOS ONE

Journal Requirements:

Reviewers' comments:

Reviewer's Responses to Questions

**Comments to the Author**

1. Is the manuscript technically sound, and do the data support the conclusions?

Reviewer #1: Partly

Reviewer #2: No

2. Has the statistical analysis been performed appropriately and rigorously? 

Reviewer #1: No

Reviewer #2: Yes

3. Have the authors made all data underlying the findings in their manuscript fully available?

Reviewer #1: No

Reviewer #2: No

4. Is the manuscript presented in an intelligible fashion and written in standard English?

Reviewer #1: Yes

Reviewer #2: Yes

5. Review Comments to the Author

Reviewer #1: I have the opportunity to review the Najjar, et al. paper titled “Role of transoesophageal echocardiography in detecting patent foramen ovale in young stroke patients: A retrospective study”. I would like to thank this honor.

This paper has valuable data, although with the inherent limitations of a retrospective analysis, especially about the bias that could indicate TOE in a given patient. Nonetheless, I have some comments.

COMMENTS

Title

-Although etiologies of ischemic strokes occurring in people aged <60 years have important differences when compared with older people, the title “young stroke patient” can be considered misleading, since the “traditional” cut-off for defining a young person is usually somewhere between 40-50 years. I would advise to be more descriptive by saying something like “…foramen ovale in stroke patients aged <60 years: A retrospective…”. This may be important, since the proportion of patients aged 51-59 was 55% in this data set. It is true that some other authors have previously used the term “young” for people aged >60 years (even in RCTs), but it has been arguable in the “strokeologists” community, given the current WHO definitions.

Abstract

-It would be interesting to declare the other TTE and TOE findings, to make possible calculating the proportion of PFOs among the other putative cardioembolic stroke etiologies. This may be important, since PFO in this age group may be an important putative etiology.

-The sentence “The rate of PFO detection by TOE was correlated with younger age” needs statistical support, since correlation (not confounding with “association”) is a specific type of statistical analysis (i.e., either Pearson’s or Spearman’s).

-Since PFO is mainly (although not solely) associated with AIS through paradoxical embolism, it is pertinent t ask, How many venous thrombotic events were identified among the whole cohort and among “cryptogenic” of “Known cause” patients? This can be deduce from the manuscript body, but it may be advisable to include this important information in the Abstract.

-In the Abstract’s conclusion is better to say “In stroke patients aged 18–60 years…”.

The statement “the effectiveness of PFO closure has been established 29 over medical treatment in reducing future reoccurrence of ischaemic stroke” lacks of supports in Results/Findings, i.e., it is not possible to conclude in such a sentence with the support of findings provided in the Abstract.

-The authors provide results such as NNT in conclusions, but not in Results. This is a little odd, since conclusions is the part of the Abstract where authors refer to findings provided ins Results section.

-I agree with the authors in that TOE may be better than TTE to detect PFO, but no findings where presented in the Abstract that forcefully support this notion.

-In brief, more results in Abstract may make this paper more citable.

Methods

-It is necessary to declare whether in this medical center some stroke patients are admitted outside the stroke unit, and if so, how this characteristic may bias the clinical practice in searching for AIS etiology.

-It is not completely clear why some patients received either TTE or TOE. Did some patients received firstly TTE and later TOE to actively find a PFO? What were the factors used in clinical practice to decide between TTE or TOE in this center?

-The etiology categories are not traditional. I would rather recommend to use TOAST or ASCOD classification to comply with “traditional” language or understanding. After this, the “new” classification proposed by the authors can also be used to make the authors’ point.

Results

-I would rather change “TTE and TOE rates” to “TTE and TOE usage rates”.

-The figure 2 can be eliminated, since only represents proportions that can be included only as text.

-The figure 3 has a legend that seems confusing. ROCs plot sensitivity Vs 1-specificity of a diagnostic tool. This ROC graphic has been used by the authors plotting age as it were a diagnostic tool, but in the Results authors state that this plot is about the performance of TTE and TOE, but this is not the case. TTE or TOE provides either a “negative” (i.e., 0) or positive (i.e., 1) test result that would produce a different ROC pattern. I would not use age as a predicting/diagnosting factor to detect PFO.

-The authors are not providing actual diagnostic performance data (i.e., sens, spec, PPV, NPV, LR+, LR-, accuracy, Youden’s index, etc.) to make the point that TOE performed better than TTE in detecting PFOs. This simply was not demonstrated in this study, so that claim cannot be supported.

Discussion

-A limitations section is needed for better interpretation of this hypothesis-generating study, especially with respect to the inherent bias of a retrospective study, and an analysis or explanation about what determined too practice either TTE or TOE in this set of patients.

-To better answer the scientific question about which is better, TTE or TOE, the best way to do it is to use TTE AND TOE in ALL patients. If some patients received either TTE or TOE, the performance of either imaging technique may be influenced by patients’ factors. Please discussed more on that.

-Please comment more about the very low frequency of deep vein thrombosis as compared with PFO, and the limitation of assigning causality by only find a PFO without active or strong identification of paradoxical embolism.

Reviewer #2: I have the opportunity to review the Najjar, et al. paper titled “Role of transoesophageal echocardiography in detecting patent foramen ovale in young stroke patients: A retrospective study”. I would like to thank this honor.

This paper has valuable data, although with the inherent limitations of a retrospective analysis, especially about the bias that could indicate TOE in a given patient. Nonetheless, I have some comments.

COMMENTS

Title

-Although etiologies of ischemic strokes occurring in people aged <60 years have important differences when compared with older people, the title “young stroke patient” can be considered misleading, since the “traditional” cut-off for defining a young person is usually somewhere between 40-50 years. I would advise to be more descriptive by saying something like “…foramen ovale in stroke patients aged <60 years: A retrospective…”. This may be important, since the proportion of patients aged 51-59 was 55% in this data set. It is true that some other authors have previously used the term “young” for people aged >60 years (even in RCTs), but it has been arguable in the “strokeologists” community, given the current WHO definitions.

Abstract

-It would be interesting to declare the other TTE and TOE findings, to make possible calculating the proportion of PFOs among the other putative cardioembolic stroke etiologies. This may be important, since PFO in this age group may be an important putative etiology.

-The sentence “The rate of PFO detection by TOE was correlated with younger age” needs statistical support, since correlation (not confounding with “association”) is a specific type of statistical analysis (i.e., either Pearson’s or Spearman’s).

-Since PFO is mainly (although not solely) associated with AIS through paradoxical embolism, it is pertinent t ask, How many venous thrombotic events were identified among the whole cohort and among “cryptogenic” of “Known cause” patients? This can be deduce from the manuscript body, but it may be advisable to include this important information in the Abstract.

-In the Abstract’s conclusion is better to say “In stroke patients aged 18–60 years…”.

The statement “the effectiveness of PFO closure has been established 29 over medical treatment in reducing future reoccurrence of ischaemic stroke” lacks of supports in Results/Findings, i.e., it is not possible to conclude in such a sentence with the support of findings provided in the Abstract.

-The authors provide results such as NNT in conclusions, but not in Results. This is a little odd, since conclusions is the part of the Abstract where authors refer to findings provided ins Results section.

-I agree with the authors in that TOE may be better than TTE to detect PFO, but no findings where presented in the Abstract that forcefully support this notion.

-In brief, more results in Abstract may make this paper more citable.

Methods

-It is necessary to declare whether in this medical center some stroke patients are admitted outside the stroke unit, and if so, how this characteristic may bias the clinical practice in searching for AIS etiology.

-It is not completely clear why some patients received either TTE or TOE. Did some patients received firstly TTE and later TOE to actively find a PFO? What were the factors used in clinical practice to decide between TTE or TOE in this center?

-The etiology categories are not traditional. I would rather recommend to use TOAST or ASCOD classification to comply with “traditional” language or understanding. After this, the “new” classification proposed by the authors can also be used to make the authors’ point.

Results

-I would rather change “TTE and TOE rates” to “TTE and TOE usage rates”.

-The figure 2 can be eliminated, since only represents proportions that can be included only as text.

-The figure 3 has a legend that seems confusing. ROCs plot sensitivity Vs 1-specificity of a diagnostic tool. This ROC graphic has been used by the authors plotting age as it were a diagnostic tool, but in the Results authors state that this plot is about the performance of TTE and TOE, but this is not the case. TTE or TOE provides either a “negative” (i.e., 0) or positive (i.e., 1) test result that would produce a different ROC pattern. I would not use age as a predicting/diagnosting factor to detect PFO.

-The authors are not providing actual diagnostic performance data (i.e., sens, spec, PPV, NPV, LR+, LR-, accuracy, Youden’s index, etc.) to make the point that TOE performed better than TTE in detecting PFOs. This simply was not demonstrated in this study, so that claim cannot be supported.

Discussion

-A limitations section is needed for better interpretation of this hypothesis-generating study, especially with respect to the inherent bias of a retrospective study, and an analysis or explanation about what determined too practice either TTE or TOE in this set of patients.

-To better answer the scientific question about which is better, TTE or TOE, the best way to do it is to use TTE AND TOE in ALL patients. If some patients received either TTE or TOE, the performance of either imaging technique may be influenced by patients’ factors. Please discussed more on that.

-Please comment more about the very low frequency of deep vein thrombosis as compared with PFO, and the limitation of assigning causality by only find a PFO without active or strong identification of paradoxical embolism.

6. PLOS authors have the option to publish the peer review history of their article (what does this mean?). If published, this will include your full peer review and any attached files.

Reviewer #1: No

Reviewer #2: No

---

## [Author Response · Author response to Decision Letter 0]

30 Sep 2020

Responses to the Reviewer’s Comments

Title

Comment 1: Although etiologies of ischemic strokes occurring in people aged <60 years have important differences when compared with older people, the title “young stroke patient” can be considered misleading, since the “traditional” cut-off for deﬁning a young person is usually somewhere between 40-50 years. I would advise to be more descriptive by saying something like “… foramen ovale in stroke patients aged <60 years: A retrospective…”. This may be important, since the proportion of patients aged 51-59 was 55% in this data set. It is true that some other authors have previously used the term “young” for people aged >60 years (even in RCTs), but it has been arguable in the “strokeologists” community, given the current WHO deﬁnitions.

Response: The title has been changed in accordance with your comment (page 1, lines 1-3).

Abstract

Comment 2: It would be interesting to declare the other TTE and TOE ﬁndings, to make possible calculating the proportion of PFOs among the other putative cardioembolic stroke etiologies. This may be important, since PFO in this age group may be an important putative etiology.

Response: ASD findings have been provided in the Results section of the Abstract (page 2, line 27).

Comment 3: The sentence “The rate of PFO detection by TOE was correlated with younger age” needs statistical support, since correlation (not confounding with “association”) is a speciﬁc type of statistical analysis (i.e., either Pearson’s or Spearman’s).

Response: In this sentence, ‘correlation’ has been changed to ‘association’ (page 2, lines 32).

Comment 4: Since PFO is mainly (although not solely) associated with AIS through paradoxical embolism, it is pertinent t ask, How many venous thrombotic events were identiﬁed among the whole cohort and among “cryptogenic” of “Known cause” patients? This can be deduce from the manuscript body, but it may be advisable to include this important information in the Abstract.

Response: This important information has been included in the Abstract (page 2, lines 26-28).

Comment 5: In the Abstract’s conclusion is better to say “In stroke patients aged 18–60 years…”.

Response: Your suggested change has been made in the Abstract (page 2, line 32).

Comment 6: The statement “the effectiveness of PFO closure has been established 29 over medical treatment in reducing future reoccurrence of ischaemic stroke” lacks of supports in Results/Findings, i.e., it is not possible to conclude in such a sentence with the support of ﬁndings provided in the Abstract.

Response: The conclusion of the Abstract has been changed to better reflect the study’s results/findings (page 2, lines 32-35).

Comment 7: The authors provide results such as NNT in conclusions, but not in Results. This is a little odd, since conclusions is the part of the Abstract where authors refer to ﬁndings provided ins Results section.

Response: The NNT results have been moved from the Conclusions to the Results (page 2, lines 30-31).

Comment 8: I agree with the authors in that TOE may be better than TTE to detect PFO, but no ﬁndings where presented in the Abstract that forcefully support this notion.

Response: Further findings have been added in the Abstract in order to support TOE > TTE (page 2, lines 28-30).

Comment 9: In brief, more results in Abstract may make this paper more citable.

Response: More results have been added to the Abstract to make this paper more citable (page 2, lines 25-31).

Methods

Comment 10: It is necessary to declare whether in this medical center some stroke patients are admitted outside the stroke unit, and if so, how this characteristic may bias the clinical practice in searching for AIS etiology.

Response: This change has been added to reflect that all patients in the centre presenting with a stroke were admitted to the stroke unit (page 4, line 128).

Comment 11: It is not completely clear why some patients received either TTE or TOE. Did some patients received ﬁrstly TTE and later TOE to actively ﬁnd a PFO? What were the factors used in clinical practice to decide between TTE or TOE in this center?

Response: This information has been added to elaborate on the choice of TTE or TOE (pages 5-6, lines 179-198).

Comment 12: The etiology categories are not traditional. I would rather recommend to use TOAST or ASCOD classiﬁcation to comply with “traditional” language or understanding. After this, the “new” classiﬁcation proposed by the authors can also be used to make the authors’ point.

Response: The traditional TOAST classification has been used along with our simplified version to address our point (page 5, lines 169-173).

Results

Comment 13: I would rather change “TTE and TOE rates” to “TTE and TOE usage rates”.

Response: Your suggested change has been made (pages 8-9, lines 262, 272, 274-276, 280).

Comment 14: The ﬁgure 2 can be eliminated, since only represents proportions that can be included only as text.

Response: The pie chart used for stroke aetiology proportions has been deleted.

Comment 15: The ﬁgure 3 has a legend that seems confusing. ROCs plot sensitivity Vs 1-speciﬁcity of a diagnostic tool. This ROC graphic has been used by the authors plotting age as it were a diagnostic tool, but in the Results authors state that this plot is about the performance of TTE and TOE, but this is not the case. TTE or TOE provides either a “negative” (i.e., 0) or positive (i.e.,1) test result that would produce a different ROC pattern. I would not use age as a predicting/diagnosting factor to detect PFO.

Response: Thank you for pointing this out. The ROC curve has been removed.

Comment 16: The authors are not providing actual diagnostic performance data (i.e., sens, spec, PPV, NPV, LR+, LR-, accuracy, Youden’s index, etc.) to make the point that TOE performed better than TTE in detecting PFOs. This simply was not demonstrated in this study, so that claim cannot be supported.

Response: The diagnostic performance data, sensitivity, specificity, predictive values, and likelihood ratios, have been provided in a new table (Table 3) for comparison between TTE and TOE.

Discussion

Comment 17: A limitations section is needed for better interpretation of this hypothesis-generating study, especially with respect to the inherent bias of a retrospective study, and an analysis or explanation about what determined too practice either TTE or TOE in this set of patients.

Response: The limitations section has been expanded and elaborated on to further reflect the weaknesses of this retrospective study (page 14, lines 503-514).

Comment 18: To better answer the scientiﬁc question about which is better, TTE or TOE, the best way to do it is to use TTE AND TOE in ALL patients. If some patients received either TTE or TOE, the performance of either imaging technique may be inﬂuenced by patients’ factors. Please discussed more on that.

Response: We have discussed this scientific question and mentioned the use of TTE and TOE in all patients (page 13, lines 483-487).

Comment 19: Please comment more about the very low frequency of deep vein thrombosis as compared with PFO, and the limitation of assigning causality by only ﬁnd a PFO without active or strong identiﬁcation of paradoxical embolism.

Response: We have addressed DVT vs. PFO and the limitation in assigning causality (page 14, lines 507-510).

---

## [Decision Letter · Decision Letter 1]

26 Oct 2020

PONE-D-20-21340R1

Role of transoesophageal echocardiography in detecting patent foramen ovale in stroke patients aged ≤60 years: A retrospective study

PLOS ONE

Dear Dr. Najjar,

Thank you for submitting your manuscript to PLOS ONE. After careful consideration, we feel that it has merit but does not fully meet PLOS ONE’s publication criteria as it currently stands. Therefore, we invite you to submit a revised version of the manuscript that addresses the points raised during the review process.

The manuscript improved substantially, with only minor details to be addressed, please refer to reviewer 1 comments on the website.

We look forward to receiving your revised manuscript.

Kind regards,

Miguel A. Barboza, MD, MSc

Academic Editor

PLOS ONE

Reviewers' comments:

Reviewer's Responses to Questions

**Comments to the Author**

1. If the authors have adequately addressed your comments raised in a previous round of review and you feel that this manuscript is now acceptable for publication, you may indicate that here to bypass the “Comments to the Author” section, enter your conflict of interest statement in the “Confidential to Editor” section, and submit your "Accept" recommendation.

Reviewer #2: All comments have been addressed

2. Is the manuscript technically sound, and do the data support the conclusions?

Reviewer #2: Yes

3. Has the statistical analysis been performed appropriately and rigorously? 

Reviewer #2: Yes

4. Have the authors made all data underlying the findings in their manuscript fully available?

Reviewer #2: Yes

5. Is the manuscript presented in an intelligible fashion and written in standard English?

Reviewer #2: Yes

6. Review Comments to the Author

Reviewer #2: I am very grateful to the authors for having listened to my recommendations and for taking the time to change the manuscript accordingly. There are only a few comments remaining:

ABSTRACT

COMMENT 1: -In the new Abstract the authors state that “The number needed to treat (NNT) to prevent the occurrence of an ischaemic stroke through PFO closure was 30.” However, since this is an observational analysis, real NNTs can be only inferred from RCTs, but not direct calculation can be obtained since no actual PFO closure was performed and compared with no closure to obtain real NNTs. Nonetheless, this calculation can be useful to the readers as a hypothetical estimation. Maybe the authors can lower the tone of this claim by saying that NNT to prevent a putative AIS recurrence through PFO closure was estimated at 30.

COMMENT 2: -There is no need to declare the NNT acronym in Abstract if it will not be used again in the text.

COMMENT 3: -In the previous Comment 4 (about the suggestion of including how many systemic venous thromboembolic events were identified in these patients) the authors responded “This important information has been included in the Abstract (page 2, lines 26-28).”, however, no information about the number of venous thrombotic events can be read from this new version of the Abstract. This may be of interest to the readers, because PFO is putatively associated with AIS through paradoxical embolism arising from thrombosis in the venous side of the circulation.

RESULTS

COMMENT 4: -I would recommend that the new Table 3 include the 95% CI for the estimations. This can be rapidly and freely obtained by using the following web-based tool:

https://ebm-tools.knowledgetranslation.net/calculator/diagnostic/

7. PLOS authors have the option to publish the peer review history of their article (what does this mean?). If published, this will include your full peer review and any attached files.

Reviewer #2: **Yes: **Erwin Chiquete, MD, PhD

---

## [Author Response · Author response to Decision Letter 1]

31 Oct 2020

Responses to the Reviewer’s Comments

ABSTRACT

COMMENT 1: -In the new Abstract the authors state that “The number needed to treat (NNT) to prevent the occurrence of an ischaemic stroke through PFO closure was 30.” However, since this is an observational analysis, real NNTs can be only inferred from RCTs, but not direct calculation can be obtained since no actual PFO closure was performed and compared with no closure to obtain real NNTs. Nonetheless, this calculation can be useful to the readers as a hypothetical estimation. Maybe the authors can lower the tone of this claim by saying that NNT to prevent a putative AIS recurrence through PFO closure was estimated at 30.

Response: The claim has been adjusted to reflect the hypothetical estimation (page 2, line 31). 

COMMENT 2: -There is no need to declare the NNT acronym in Abstract if it will not be used again in the text.

Response: The acronym has been removed (page 2, line 30).

COMMENT 3: -In the previous Comment 4 (about the suggestion of including how many systemic venous thromboembolic events were identified in these patients) the authors responded “This important information has been included in the Abstract (page 2, lines 26-28).”, however, no information about the number of venous thrombotic events can be read from this new version of the Abstract. This may be of interest to the readers, because PFO is putatively associated with AIS through paradoxical embolism arising from thrombosis in the venous side of the circulation.

Response: Apologies, the number of venous thromboembolic events have been added (page 2, lines 26-27).

RESULTS

COMMENT 4: -I would recommend that the new Table 3 include the 95% CI for the estimations. This can be rapidly and freely obtained by using the following web-based tool:

https://ebm-tools.knowledgetranslation.net/calculator/diagnostic/

Response: The 95% CI for the estimations were added in table 3 (page 10, lines 214-218).

---

## [Decision Letter · Decision Letter 2]

11 Nov 2020

Role of transoesophageal echocardiography in detecting patent foramen ovale in stroke patients aged ≤60 years: A retrospective study

PONE-D-20-21340R2

Dear Dr. Najjar,

We’re pleased to inform you that your manuscript has been judged scientifically suitable for publication and will be formally accepted for publication once it meets all outstanding technical requirements.

Kind regards,

Miguel A. Barboza, MD, MSc

Academic Editor

PLOS ONE

Additional Editor Comments (optional):

Reviewers' comments:

Reviewer's Responses to Questions

**Comments to the Author**

1. If the authors have adequately addressed your comments raised in a previous round of review and you feel that this manuscript is now acceptable for publication, you may indicate that here to bypass the “Comments to the Author” section, enter your conflict of interest statement in the “Confidential to Editor” section, and submit your "Accept" recommendation.

Reviewer #2: All comments have been addressed

2. Is the manuscript technically sound, and do the data support the conclusions?

Reviewer #2: Yes

3. Has the statistical analysis been performed appropriately and rigorously? 

Reviewer #2: Yes

4. Have the authors made all data underlying the findings in their manuscript fully available?

Reviewer #2: Yes

5. Is the manuscript presented in an intelligible fashion and written in standard English?

Reviewer #2: Yes

6. Review Comments to the Author

Reviewer #2: Thank you. The authirs have adderessed my previous inquiries and the paper has improved very substantially.

7. PLOS authors have the option to publish the peer review history of their article (what does this mean?). If published, this will include your full peer review and any attached files.

Reviewer #2: **Yes: **Erwin Chiquete, MD, PhD

---

## [Editor Report · Acceptance letter]

16 Nov 2020

PONE-D-20-21340R2 

Role of transoesophageal echocardiography in detecting patent foramen ovale in stroke patients aged ≤60 years: A retrospective study 

Dear Dr. Najjar:

I'm pleased to inform you that your manuscript has been deemed suitable for publication in PLOS ONE. Congratulations! Your manuscript is now with our production department. 

Kind regards, 

on behalf of

Dr. Miguel A. Barboza 

Academic Editor

PLOS ONE